# Fast Localization and Characterization of Underground Targets with a Towed Transient Electromagnetic Array System

**DOI:** 10.3390/s22041648

**Published:** 2022-02-20

**Authors:** Lijie Wang, Shuang Zhang, Shudong Chen, Chaopeng Luo

**Affiliations:** 1College of Electronic Science and Engineering, Jilin University, Changchun 130012, China; lijiew20@mails.jlu.edu.cn (L.W.); zhangshuang@jlu.edu.cn (S.Z.); 2Science and Technology on Near-Surface Detection Laboratory, Wuxi 214035, China

**Keywords:** unexploded ordnance (UXO), transient electromagnetic array system, magnetic gradient tensor, singular value decomposition (SVD)

## Abstract

A fast inversion algorithm combined with the transient electromagnetic (TEM) detection system has important significance for improving the detection efficiency of unexploded ordnance. The traditional algorithms, such as differential evolution or Gauss–Newton algorithms, usually require tens to thousands of iterations to locate the underground target. A new algorithm with a magnetic gradient tensor and singular value decomposition (SVD) to estimate the target position and characterization quickly and accurately is proposed in this paper. Two modes of magnetic gradient tensor are constructed to accurately locate shallow and deep targets, respectively. The SVD algorithm is applied to the responses to estimate the electromagnetic characteristics of the target quickly and accurately. To verify the performance of the proposed algorithm, a towed TEM sensor is designed, which is constructed with three transmitting coils and nine three-component receiving coils arranged in a 3 × 3 array. Field experiments in survey and cued modes were taken to verify the performance of the proposed algorithm and the towed system. Results show that the magnetic gradient tensor algorithm proposed in this paper can accurately locate a single target within 2.0 m depth, and the error of depth is no more than 8 cm. Even for overlapping response of multi targets, the error of depth is no more than 12 cm. The underground target can be accurately characterized by the SVD algorithm. For targets with depths over 2.0 m, the signal-to-noise ratio of characteristic response estimated by SVD is higher than that of the traditional method. The proposed method needs approximately 40 ms, only 1% of the traditional one, considerably improving detection efficiency and laying a theoretical and experimental foundation for real-time data processing.

## 1. Introduction

Unexploded ordnance (UXO) pollution has caused global concern. It threatens human health and hinders economic construction and reuse of land on which humans depend [1,2]. Metallic shrapnel and debris are usually scattered around UXO, making it more difficult and inefficient to clean and identify [3,4]. Therefore, detecting and identifying UXOs quickly and accurately is particularly important.

In recent decades, a variety of detection methods such as magnetic detection [5,6], ground-penetrating radar (GPR) [7,8], and electromagnetic induction (EMI) [9,10] have been developed. Magnetic detection is a passive detection method based on the abnormal detection of targets in the geomagnetic field, which is a fast, convenient, and efficient method. GPR transmits high-frequency electromagnetic waves for target detection, which is used for target imaging and is susceptible to geological conditions. The EMI method includes the frequency [11,12] and time domains [13,14], which is an active detection method, and the working frequency ranges from tens to hundreds of kHz. Compared with the previous two detection methods, the EMI method is less affected by geology and has a strong anti-interference ability. The time-domain EMI method or transient electromagnetic method (TEM) has been effectively applied for underground target detection.

Detection, inversion, and classification are the key steps in the cleanup of UXOs. In detection, a series of TEM systems are designed for UXOs detection based on the TEM system. The most typical systems include two categories: portable TEM systems and vehicle TEM systems. Portable sensors include the Man-Portable Vector MPV-I and MPV-II designed by G & G Sciences [15,16], and a portable metal detector designed by Jilin University [17]. Vehicle systems include the Berkeley UXO Discriminator [18], Metal-Mapper designed by Geometrics [19], and Time-domain Electromagnetic Multi-sensor Tower Array Detection System [20]. Unlike portable systems, the vehicle systems have a larger transmitting magnetic moment, deeper detection range, and higher detection efficiency. In addition, the vehicle systems adopt multiple transmitters and receivers and uses single measurement to achieve target location and characterization. These systems are widely used to detect targets in flat and large areas, and they can achieve rapid positioning and characterization of underground targets. In classification, various algorithms like support vector machines, deep learning, and convolutional networks [21,22,23] have been used to distinguish UXOs from harmless targets. In an inversion, some iterative algorithms are usually used to locate and characterize targets, such as the Gauss–Newton [24,25] and differential evolution (DE) algorithms [26,27]. Which usually require dozens of or thousands of iterations.

Different from the iterative algorithm, the magnetic gradient tensor localization algorithm, which can directly calculate the target position, has been effectively used to locate targets in magnetic detection [28,29,30,31,32,33]. The localization accuracy with the magnetic gradient tensor is mainly affected by signal-to-noise ratio (SNR) and calculated error of the magnetic gradient tensor. Different from magnetic detection, the response amplitude of TEM detection is inversely proportional to the sixth power of the distance from the target to the sensor. Thus, when the target is far from the sensor, the SNR is very low. When the target is close to the sensor, the calculated error of magnetic gradient tensor constructed by differential response will increase rapidly. The low SNR and the calculated error will all lead to large localization errors, which limits the application of the magnetic gradient tensor in UXO detection with the TEM system. To locate the target quickly and accurately, two modes of magnetic gradient tensor are proposed with TEM detection in this paper to locate shallow and deep targets, respectively. In addition, singular value decomposition (SVD) has been applied to calculate the electromagnetic characteristics of the target with low SNR.

The remaining chapters of this paper are structured as follows. Section 2, gives a detailed introduction to the TEM system and structure of the towed sensor. Section 3, is data processing, which mainly describes the dipole model, target location, and characterization. Section 4, provides the experiment and analysis. Section 5 summarizes this paper.

## 2. Towed TEM System Description

To verify the performance of the proposed method, a towed array sensor is designed in this paper to excite the underground target from three different directions and obtain the response with nine three-component receiving coils arranged in a 3 × 3 array. The overall workflow and structure of the towed TEM system are introduced in this section.

### 2.1. System Work Procedure

The workflow of the newly designed towed TEM system is shown in Figure 1.

The towed system is controlled by a National Instruments data acquisition card USB6349 and a laptop. The USB6349 is used to generate two PWMs with a 25% duty cycle to produce the desired signal through LabVIEW. The full-bridge circuit transmits a bipolar trapezoidal wave current. The 27 channel responses of nine three-component receiving coils are simultaneously collected by USB6349 at a sampling rate of 200 kHz, which are delivered to the computer for real-time display and target discrimination. Three switches control the excitation of the three transmitting coils. Similar to the portable systems, the towed TEM array system also operates in survey and cued modes. In survey mode, the z-component transmitting coil works at a frequency of 125 Hz to roughly detect the anomalies. In cued mode, three transmitting coils transmit the current at a frequency of 12.5 Hz to excite the underground target in turn. The nine three-component receiving coils pick up the target response synchronously.

### 2.2. Towed Sensor Array

The structure and picture of the towed TEM array system with three transmitting coils and nine three-component receiving coils are shown in Figure 2.

In Figure 2a, the green, yellow, and red squares are the x-, y-, and z-component transmitting coils wound by overlapping loops with the copper line with a cross-area of 6 mm^2^, respectively. The side length and number of turns of the z-component transmitting coil are 0.95 m and 20, respectively. The x- and y-component transmitting coils are constructed with two inverted series rectangular coils. These rectangular coils are constructed with a size of 0.85 m × 0.35 m and a number of turns of 20. A total of nine three-component receiving coils are arranged in a 3 × 3 array with a 20 cm interval. The side length, resonance frequency, and distance between sections of each receiving coil are 8 cm, 230 kHz, and 7 mm, respectively. The nine receiving coils adopt double-layer shielding combined with a center-tapped grounding to improve the SNR of the receiving response. Figure 2b is the equivalent circuit. Figure 2c is a physical picture of the towed system. Table 1 shows the detailed parameters.

### 2.3. Transmitting Current and Primary Field

The transmitting currents in the survey and cued modes are collected at a sampling rate of 200 kHz by USB6349, as shown in Figure 3.

As shown in Figure 3a, in survey mode, only the z-component transmitting coil is used to emit a 125 Hz bipolar rectangular wave with a 50% duty cycle. The amplitude of the transmitting current is approximately 15 A, and the pulse width is 2 ms. Due to the capacitive energy storage technology, the front edge of the transmitting current is greatly improved, which effectively improves the excitation efficiency. In Figure 3b, in cued mode, x-, y-, and z-component transmitting coils emit a 12.5 Hz bipolar rectangular wave with a 50% duty cycle. The pulse width is 20 ms, and the transmitting current amplitudes of the x, y, and z coils are approximately 15.2, 14.6, and 16.5 A, respectively.

On the basis of the structure, parameter, and transmitting current of the sensor, the primary fields are simulated by Ansoft MAXWELL 15.0 soft in cued mode. Figure 4 is the simulation results of the primary field for the x- and z-component transmitting coils.

Figure 4a is the vector distribution of the primary field generated by the x-component transmitting coil. The primary field below the sensor only contains the x-component. That is, the target below the sensor can be fully excited in the x direction by the x-component transmitting coil. Given that the structure and parameters of the y-component transmitting coil are almost the same as those of the x coil, the target below the sensor can also be fully excited in the y direction by the y-component transmitting coil. Figure 4b is the vector distribution of the primary field excited by the z-component transmitting coil. The z-component of the primary field is the strongest below the transmitting coil, and the strength decreases with the increase in depth. Therefore, the target below the sensor can be effectively excited in the z direction by the z-component transmitting coil.

In general, when three orthogonal transmitting coils are used to excite the target in turn, it is ensured that the target is fully excited from three orthogonal directions.

## 3. Data Processing

### 3.1. Single Dipole Model

The response inversion is performed based on the single dipole model [17], as shown in Figure 5.

The transmitting coil generates a primary field that changes with time to excite the target. The secondary field ***B***_S_ generated by the induced eddy current of the target at the position of receiving coil ***r*** can be expressed by the Green’s function ***G***(***R***) and the dipole moment ***m*** as
(1)$$B_{S}=\frac{1}{4\pi R^{3}}(3n^{\prime }n-I)m=G(R)m$$
where *R* is the modulus of ***R***, and ***n*** = ***R***/*R*. The relative position ***R*** = ***r*** − ***r***_d_. ***I*** is an identity matrix. ***G***(***R***) is Green’s function, which depends on the target position ***R***. The dipole moment ***m*** is determined by the magnetic polarizability tensor ***M*** and the primary field ***B***_p_ of the target as
(2)$$m=MB_{P}(r_{d}),$$
where ***M*** depends on the shape, size, orientation, permeability, and conductivity of the target, which is a symmetric matrix.

In EMI detection, the receiving coil picks up the target response ***V***, which is the time derivative of the magnetic field ***B***_S_. According to Equations (1) and (2), the target response ***V*** can be written as [26]
(3)$$V=-G\left(R\right)\frac{dM}{dt}B_{P}\left(r_{d}\right)=G\left(R\right)L\left(t\right)B_{P}\left(r_{d}\right),$$
where ***L***(*t*) is the characteristic matrix, opposite of the time derivative of ***M***.

### 3.2. Target Localization with Magnetic Gradient Tensor

In view of the situation of the dipole forward model, this paper uses the magnetic gradient tensor method to transform the traditional nonlinear problem into a linear problem. The magnetic gradient tensor has been proven to be effective in locating targets in magnetic detection. Special sensor structures, such as cross-shaped and double-cross-shaped structures, are usually used to construct the magnetic gradient tensor. In EMI detection, the target position estimated by the gradient tensor ***G*** and the target response ***V*** is given by [34]
(4)$$R=-3G^{-1}V,$$
where ***G*** is a 3 × 3 symmetric matrix, and the sum of diagonal elements is zero, which is given by
(5)$$G=\left[\begin{array}{ccc} \partial V_{x}/\partial x & \partial V_{x}/\partial y & \partial V_{x}/\partial z\\ \partial V_{y}/\partial x & \partial V_{y}/\partial y & \partial V_{y}/\partial z\\ \partial V_{z}/\partial x & \partial V_{z}/\partial y & \partial V_{z}/\partial z \end{array}\right],$$
where ***G*** represents the rate of change of the target response ***V*** in space.

Due to the high SNR of the shallow target response, the magnetic gradient tensor is obtained by the four adjacent receiving coils, such as R1, R2, R4, R5 shown in Figure 6a, which maximizes the use of the high spatial resolution of the short-distance receiving coils, thereby considerably reducing the calculated error of the magnetic gradient ***G***.

For deep targets responses with a low SNR, the magnetic gradient tensor at the center point R5 is obtained by the receiving coils R2, R4, R6, and R8, as shown in Figure 6b. The distance between the receiving coils used to construct the magnetic gradient tensor is doubled, which significantly improves the SNR of the magnetic gradient tensor response.

In Figure 6a, ***V***_E1_ is the equivalent response in the center of the four sensors R1, R2, R4, and R5, which is written as
(6)$$V_{E1}=\frac{V_{1}+V_{2}+V_{4}+V_{5}}{4},$$
where ***V***_1_, ***V***_2_, ***V***_4_, and ***V***_5_ are target responses obtained from the receiving coils R1, R2, R4, and R5, respectively.

According to the four sensors, the five independent elements of the gradient tensor ***G*** are calculated as
(7)$$\{\begin{array}{c} \partial V_{x}/\partial x=(V_{4x}+V_{5x}-V_{1x}-V_{2x})/2d\\ \partial V_{x}/\partial y=(V_{4y}+V_{5y}-V_{1y}-V_{2y})/2d\\ \partial V_{x}/\partial z=(V_{4z}+V_{5z}-V_{1z}-V_{2z})/2d\\ \partial V_{y}/\partial y=(V_{2y}+V_{5y}-V_{1y}-V_{4y})/2d\\ \partial V_{y}/\partial z=(V_{2z}+V_{5z}-V_{1z}-V_{4z})/2d\\ \partial V_{z}/\partial z=-(\partial V_{x}/\partial x+\partial V_{y}/\partial y) \end{array},$$
where *V_αβ_* is the target response, *α* = 1, 2, 4, 5 is the number of the receiving coils, and *β* = x, y, z is three components of the receiving coil.

In Figure 6b, R5 records the target response ***V***. The five independent elements of the gradient tensor ***G*** are calculated as
(8)$$\{\begin{array}{c} \partial V_{x}/\partial x=(V_{8x}-V_{2x})/2d\\ \partial V_{x}/\partial y=(V_{8y}-V_{2y})/2d\\ \partial V_{x}/\partial z=(V_{8z}-V_{2z})/2d\\ \partial V_{y}/\partial y=(V_{6y}-V_{4y})/2d\\ \partial V_{y}/\partial z=(V_{6z}-V_{4z})/2d\\ \partial V_{z}/\partial z=-(\partial V_{x}/\partial x+\partial V_{y}/\partial y) \end{array},$$
where *V_αβ_* is the target response, *α* = 2, 4, 6, 8 is the number of the receiving coils, and *β* = x, y, z is three components of the receiving coils.

According to the magnetic gradient tensor constructed in Figure 6a, the shallow target position can be estimated by substituting Equations (5)–(7) into Equation (4). The deep target position can be estimated by substituting the response of R5 in Figure 6b and Equations (5) and (8) into Equation (4).

### 3.3. Target Characterization

The accurate and fast estimation of target characteristics has important significance for target recognition and classification. Usually, according to the observed target response and target position, the electromagnetic characteristic matrix of the target can be directly calculated [17].

If the target position ***r***_d_ is known, the electromagnetic characteristic matrix ***L***(*t*) of the target can be calculated according to Equation (3) as
(9)$$L(t)=\frac{G(R)\prime V_{\mathrm{obs}}B_{p}(r_{d})\prime }{G(R)\prime G(R)B_{p}(r_{d})B_{p}(r_{d})\prime },$$
where ***V***_obs_ is the observed target response. According to Equation (9), the target characteristic matrix ***L***(*t*) can be calculated by the observed target response ***V***_obs_, the primary field ***B***_P_ (***r***_d_), and Green’s function ***G***(***R***). The electromagnetic characteristics of the target can be obtained by the SVD of the electromagnetic matrix ***L***(*t*), which is written as
(10)$$L(t)=A\left[\begin{array}{ccc} l_{p}\left(t\right) & 0 & 0\\ 0 & l_{v1}\left(t\right) & 0\\ 0 & 0 & l_{v2}\left(t\right) \end{array}\right]A\prime ,$$
where ***A*** is the Euler rotation matrix. *l*_p_(*t*) represents the polarizability along the long axis of the target. *l*_v1_(*t*) and *l*_v2_(*t*) represent the polarizabilities perpendicular to the long axis of the target. For axisymmetric targets, *l*_v1_(*t*) and *l*_v2_(*t*) are highly consistent. The electromagnetic characteristic of the target can be denoted as ***l***(*t*), which is expressed as
(11)$$l\left(t\right)=\left[\begin{array}{ccc} l_{p}\left(t\right) & l_{v1}\left(t\right) & l_{v2}\left(t\right) \end{array}\right]$$

The traditional characterization method described in Equations (9)–(11) is based on the target position. When the response SNR is low, the target localization error will increase, resulting in an increase in the electromagnetic characteristic response error. To improve the SNR of estimated electromagnetic characteristics for targets with large depth, the SVD is applied on the obtained response data for the towed TEM array system, which does not need to estimate the target position.

According to the towed sensor in Figure 2, the target response ***V*** can be given by
(12)$$V=\left[\begin{array}{ccc} V_{1x} & V_{1y} & V_{1z}\\ \vdots & \vdots & \vdots \\ V_{9x} & V_{9y} & V_{9z} \end{array}\right]=\left[\begin{array}{c} G\left(R_{1}\right)\\ \vdots \\ G\left(R_{9}\right) \end{array}\right]L\left(t\right)\left[\begin{array}{ccc} B_{Px}\left(r_{d}\right) & B_{Py}\left(r_{d}\right) & B_{Pz}\left(r_{d}\right) \end{array}\right],$$
where ***V****_ij_*, *i* = 1…9, is the number of nine receiving coils, and *j* = *x*, *y*, *z*, is the number of three transmitting coils. The target response ***V*** is a matrix of 27 × 3. The Green’s function ***G***(***R***) and primary field ***B***_P_ (***r***_d_) only depend on the target position ***R***. The characteristic matrix ***L***(*t*) depends on the attitude and electromagnetic characteristics of the target. The SVD applied on the target response ***V*** can be given by
(13)$$V=U\left[\begin{array}{ccc} k_{p}l_{p}\left(t\right) & 0 & 0\\ 0 & k_{v1}l_{v1}\left(t\right) & 0\\ 0 & 0 & k_{v2}l_{v2}\left(t\right) \end{array}\right]S,$$
where *k*_p_, *k*_v1_, *k*_v2_ are coefficients between diagonal element and truth, representing the influence of Green’s function ***G***(***R***) and primary field ***B***_P_ (***r***_d_) on the target characterization.

In Equation (13), the target response ***V*** is decomposed by SVD. The attenuation characteristics of diagonal elements with time are exactly the same as the electromagnetic characteristics ***l***(*t*) of the target, and the only difference is amplitude. The electromagnetic characteristics estimated by the SVD need to be corrected according to the amplitude of the electromagnetic characteristics in a certain period of time estimated by the target position. Which reduces the impact of inaccurate amplitude on subsequent target recognition.

### 3.4. Detection Process

The detection and data processing with towed TEM system are shown in Figure 7.

As shown in Figure 7, the measurement and data processing can be divided into five steps. First, in survey mode, the z-component transmitting coil is used to excite the underground target at a frequency of 125 Hz, and the z-component of receiving coil R5 is used to draw the target response along the survey line. Second, the rough positions of underground targets are determined according to the maximum amplitude of target response. In cued mode, the three transmitting coils are used to excite the target in turn in these response positions, and nine receiving coils obtain the target response synchronously. The target positions are calculated by four-coil gradient tensor localization for shallow depth and five-coil gradient tensor localization for large depth. Finally, according to the estimated positions, the electromagnetic characteristics of the targets estimated by the SVD are corrected. The performance of the towed system and proposed algorithm will be verified in the field experiments.

## 4. Experiment and Analysis

Field experiments were conducted in the southern suburb of Changchun City, Jilin Province to verify the performance of the proposed sensor, the target localization, and characterization algorithms. The specific experimental process and results are analyzed in the following sections.

### 4.1. Experiment Description

A series of targets, including harmless targets and UXOs, were buried in field experiments. These targets are shown in Figure 8. The detailed parameters are shown in Table 2 and Table 3.

As shown in Figure 8, 12 types of UXOs and 9 types of harmless targets were numbered U1 to U12 and O1 to O9, respectively. In Table 2, the length of the UXOs (U1 to U12) is from 18 cm to 65 cm, and the outer diameter is from 37 mm to 130 mm. In Table 3, the lengths of the two bullet shells (O1 and O2) with outer diameters of 30 and 37 mm are 16 and 25 cm. The length of four steel pipes (O3 to O6) with an outer diameter of 75 mm is from 5 cm to 30 cm. O7 is a discus with an outer diameter of 150 mm and a thickness of 2 cm. O8 is a steel ball with a diameter of 64 mm. O9 is a three-way connector with a height of 12.5 cm.

All of these targets were buried in a 13 m × 8 m area. Figure 9 is the specific distribution of these targets.

As shown in Figure 9a, all 19 UXOs and 10 harmless targets were buried separately or with two together in the measurement area. On the y = 1 m survey line, 4 harmless targets (O1, O2, O7, and O8) were buried separately. On the y = 3 m survey line, 4 UXOs (U2, U4, U6, and U7) were buried separately. On the y = 6 m survey line, 5 UXOs (U8 to U12) were buried separately. The remaining targets were divided into 8 groups with 2 targets in each group, and the distance between the two targets was 40 cm.

On the basis of the towed TEM system, survey and cued modes were performed on these buried targets in turn to verify the performance of the inversion algorithms.

### 4.2. Experimental Results and Analysis in Survey Mode

In survey mode, the towed TEM system transmits a 125 Hz bipolar rectangular current with the z-component transmitting coil. The sensor moves along the survey line in the x direction at a speed of 1 m/s and collects a response every 0.2 s. A total of nine measurement lines with 16,038 responses were collected in the field experiment. Each line collected 66 points with 1782 responses. Finally, the response of the z-component of the receiving coil R5 was normalized with transmitting current, and the area in survey mode is shown in Figure 10.

Figure 10 shows that dozens of abnormal responses distributed in the area were detected by the towed TEM system. A total of 5 abnormal responses were observed in the survey line y = 6.0 m, and 7 abnormal responses were observed in the survey line y = 3.0 m. The abnormal responses in the survey line y = 1.0 m are relatively complex. Multiple groups of abnormal responses are superimposed on each other. On the basis of the amplitude of these abnormal responses, we can roughly determine the horizontal positions of targets. However, determining which one is UXO or which one is a harmless target is impossible. Therefore, we need to further process these responses and determine a series of measurement points from the line response for cued mode detection. The survey line responses at y = 6, y = 3, and y = 1 m are shown in Figure 11.

Figure 11 is the z component survey line responses of the receiving coil R5 at y = 6, 3, and 1 m. On the basis of these three survey lines, a series of measurement points are selected for cued mode. For the survey line at y = 6 m, the amplitude of target response is the smallest. The 5 positions x = 0.8, 4.0, 7.0, 10.2, and 12.4 m, which are the local maximums of survey line response, were chosen for cued mode detection. For the survey line at y = 3.0 m, the amplitude of the target response is much greater. The 7 positions at x = 1.2, 3.2, 5.0, 7.0, 9.2, 11.0, and 12.0 m, local maximums of survey line response, were chosen for cued mode detection. For the survey line response at y = 1.0 m, the amplitude of target response is the largest. The 10 positions at x = 0.8, 2.2, 4.0, 5.2, 6.0, 7.0, 8.4, 10.0, 11.2, and 12.0 m, local maximums of survey line response, were chosen for cued mode detection.

In general, in survey mode, the towed system can detect all targets within 2.1 m depth and can roughly determine the horizontal position of the underground target, providing a horizontal position for cued detection.

### 4.3. Experimental Results and Analysis in Cued Mode

According to the 22 maximum points determined in survey mode, detection in cued mode is carried out for each point. The towed system first moves over these determined positions, and then excites the underground targets with three transmitting coils in turn. Nine three-component receiving coils obtain the target response simultaneously. With these responses, the target localization and characterization can be realized. In this section, the measurement points with local maximum amplitude on the three survey lines (y = 6, 3, and 1 m) are detected in cued mode, and the results are discussed according to the survey lines.

#### 4.3.1. Inversion Results at y = 6 m

The cued mode detection was carried out at positions where y = 6.0 m, x = 0.8, 4.0, 7.0, 10.2, and 12.4 m. Target positions were estimated with a magnetic gradient tensor constructed with five coils in Figure 6b and compared with the truth position. The running time of the proposed method is also compared with the traditional DE algorithm. The target localization results are shown in Table 4.

As shown in Table 4, for a target with a depth of more than 2 m, the horizontal localization error is large, reaching 25 cm. The z-component has a relatively large response and higher SNR, and its error is small, with a maximum error of only 13 cm. As the depth decreases, the SNR of the x and y component responses increases rapidly, and the errors in horizontal positions and depth do not exceed 11 and 5 cm, respectively. In general, the error can be ignored compared with the target depth. The localization result of the gradient tensor is close to the truth position of the target. The proposed method takes approximately 40 ms to achieve target location and characterization, which is only approximately 1% of the traditional method.

With the estimated positions, the target electromagnetic characteristics estimated by the SVD were corrected. The target characteristic response was sampled from 0.2 ms to 20 ms. Both calculated and corrected electromagnetic characteristics are compared in Figure 12.

The characteristic responses *l*_p_(*t*) estimated by the SVD and target position, respectively, are shown in Figure 12. In Figure 12a,b, due to the large depth (more than 2.0 m), the SNR of characteristic response *l*_p_(*t*) calculated by target position is very low. The SNR of characteristic response *l*_p_(*t*) estimated by the SVD is much higher than that produced with the traditional method. In Figure 12c, when the target depth decreases to approximately 1.7 m, the characteristic response estimated by two methods fits well with that corrected by SVD after 0.6 ms. As the target depth decreases to 1.3 and 1.5 m, as shown in Figure 12d,e, the characteristic responses *l*_p_(*t*) estimated by the two methods are highly consistent with high SNR.

In general, when the target depth is deeper, the characteristic response *l*_p_(*t*) estimated by SVD has a higher SNR. As the target depth decreases, the characteristic responses estimated with two methods are highly consistent.

#### 4.3.2. Inversion Results at y = 3 m

With the gradient tensor localization method described in Figure 6b, the positions of 7 abnormal responses at y = 3.0 m, x = 1.2, 3.2, 5.0, 7.0, 9.2, 11.0, and 12.0 m were calculated, and the target localization results are shown in Table 5.

As shown in Table 5, for a single target with a depth of approximately 1 m, the errors of horizontal positions and depth do not exceed 9 and 6 cm, respectively. For multi-targets, due to response overlapping, the localization error increases. The errors of the horizontal positions and depth are no more than 10 and 7 cm, respectively, except for U4. The localization result is biased toward the target with a strong response, so the localization error of U4 reaches 14 and 12 cm in the horizontal direction and depth. In general, the proposed method can quickly and accurately locate the target with a depth of approximately 1 m.

On the survey line y = 3.0 m, the vertical axis characteristic responses of the target estimated by the SVD and target position, respectively, are shown in Figure 13.

As shown in Figure 13b,c,f,g, for a single target, the characteristic responses *l*_p_(*t*) estimated by two methods are highly consistent and have high SNR. For multi-targets, shown in Figure 13a,d,e, the estimated characteristic responses with SVD are influenced by adjacent targets and determined by the one with a larger response. In Figure 13a, the estimated characteristic responses with SVD are determined by target one early and target two later; the same applies for Figure 13e. For two identical targets, the estimated characteristic responses with two methods are consistent.

Overall, for a single target, the characteristic response of the target estimated by two methods is highly consistent with high SNR. For multi-targets, the characteristic response of each target can also be effectively estimated by the target position.

#### 4.3.3. Inversion Results at y = 1 m

In this section, the gradient tensor in Figure 6a is used to locate a target, a total of 9 measurement points y = 6.0 m, x = 0.8, 2.2, 4.0, 5.2, 6.0, 7.0, 8.4, 10.0, 11.2, and 12.0 m. The target localization results are shown in Table 6.

As shown in Table 6, the target location and characterization take approximately 40 ms, which is approximately 1% of the traditional one. Most of the target depths are approximately 50 cm, with high SNR. Therefore, the localization method in Figure 6a can effectively reduce the calculated error of magnetic gradient tensor and locate targets precisely. For the remaining targets, even for two targets with overlapping signals, the depth error is no more than 7 cm. Owing to close distance, the responses overlap each other, and the horizontal position error is relatively large, reaching 14 cm. The target O3 cannot be detected because its signal is much smaller than the adjacent target O9.

On the basis of the above analysis, when the target depth is less than 1 m, the target characteristic response estimated by the target position has a high SNR. Therefore, in the survey line y = 1 m, only the characteristic response based on the target position is calculated, and the results are shown in Figure 14.

For a single target, as shown in Figure 14d,f,h, the estimated characteristic response *l*_p_(*t*) decays slowly and is slightly higher than the characteristic response *l*_v1_*(t*) and *l*_v2_(*t*). In Figure 14a, the characteristic responses *l*_v1_*(t*) and *l*_v2_(*t*) perpendicular to the target axis of the discus are highly coincident and slightly higher than the characteristic response *l*_p_(*t*). In Figure 14e, the characteristic responses *l*_p_(*t*), *l*_v1_*(t*), and *l*_v2_(*t*) of the steel ball are highly consistent and have a high SNR. In Figure 14f, due to the influence of adjacent targets, the characteristic responses *l*_v1_*(t*) and *l*_v2_(*t*) of the bullet shell have a significant difference. In Figure 14h, the characteristic responses of the three-way connector are different. The localization of a 5 cm iron pipe is failed due to the influence of the nearby target, resulting in the failure of characterization.

For multi-targets, as shown in Figure 14b,c,g,i, the target characterization can be realized. In Figure 14c,g,i, the amplitude difference of UXOs and harmless targets is small, within 1 ms. In the late stage, the characteristic responses of the UXOs are higher than that of harmless targets. Therefore, the harmless targets can be effectively separated from UXOs with the late characteristic response.

In summary, when the target depth is deep (over 1.6 m), SVD can greatly improve the SNR of the characteristic response. When the target depth is shallow (no more than 1.6 m), SVD and the target position can precisely estimate the characteristic response of a single target. For multi-targets, the characteristic response estimated by SVD or target position will be affected by the adjacent targets. The characteristic response can be effectively estimated by the target position, and UXOs and harmless targets can also be effectively distinguished. The magnetic gradient tensor constructed by the towed system can locate and characterize the target quickly and accurately. It takes only approximately 1% of the traditional method, significantly improving the detection performance, thereby providing a basis for fast classification and positioning of targets.

## 5. Conclusions

In this paper, a towed TEM system consisting of three transmitting coils and nine receiving coils is designed. According to the obtained responses, two forms of magnetic gradient tensors are constructed to quickly and accurately locate targets at different depths. According to the target position and SVD, the target characterization at different depths can be estimated accurately.

The towed system is composed of three transmitting coils and nine three-component receiving coils in a 3 × 3 array. The x- and y-component transmitting coils are a pair of coplanar and reverse-winding rectangular coils, with a side length of 0.85 m × 0.35 m. The side length of the square z-component transmitting coil is 0.96 m. The simulation results show that the primary fields under the detection system generated by the x-, y-, and z-component transmitting coils are orthogonal, exciting the target from three directions.

In this paper, the magnetic gradient tensor localization of magnetic detection is applied to TEM UXO detection, enabling the quick location of the underground target without iteration. For the two problems of large error of magnetic gradient tensor in shallow and low SNR in deep, two types of magnetic gradient tensors, namely, rectangular and cross-shaped, were constructed to reduce the error and improve the SNR. Experimental results show that, for a single target in depth (less than 2 m), the localization error of depth is less than 8 cm; for multiple targets, the error of depth is no more than 12 cm. As the target depth increases (over 2 m), the SNR decreases, and the localization error increases. Results show that the two magnetic gradient methods constructed in this paper can quickly and effectively estimate the target position in deep and shallow locations.

Experimental results show that for a target with a depth of more than 1.6 m, the characteristic response of the target estimated by SVD has high SNR. For the target with a depth of less than 1.6 m, the electromagnetic response can be accurately calculated by the localization results of the magnetic gradient tensor. According to the target characterization, the shape of the target can be easily distinguished.

Compared with the traditional method, the method proposed in this paper still has shortcomings in multi-target detection. However, the magnetic gradient tensor localization method proposed here does not require iteration, and the running time is only 1% of the traditional method, considerably improving detection efficiency and providing a possibility for real-time detection and processing of UXO. In the future, the proposed method can be combined with traditional detection methods to deal with more complex situations in TEM and realize fast and accurate detection of targets.

## Figures and Tables

**Figure 1 sensors-22-01648-f001:**
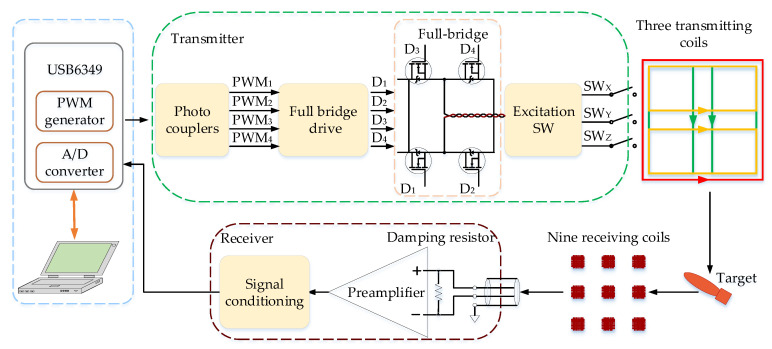
Schematic diagram of the towed system.

**Figure 2 sensors-22-01648-f002:**
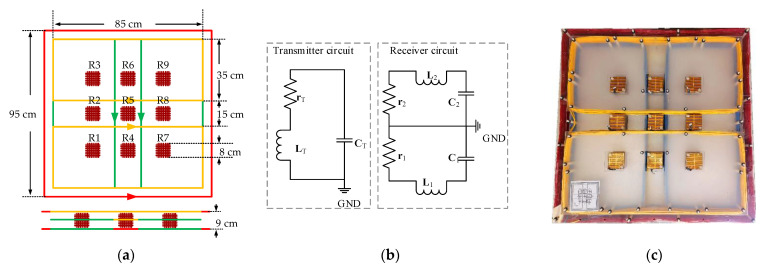
(**a**) Structure of towed TEM sensor array; (**b**) equivalent circuit diagram; (**c**) physical picture of the towed sensor.

**Figure 3 sensors-22-01648-f003:**
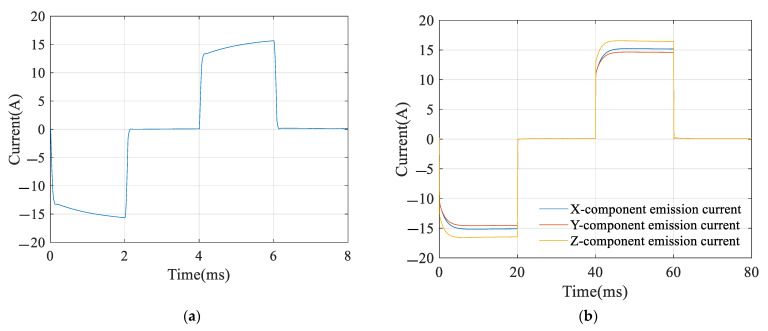
(**a**) Transmitting current in survey mode; (**b**) transmitting current in cued mode.

**Figure 4 sensors-22-01648-f004:**
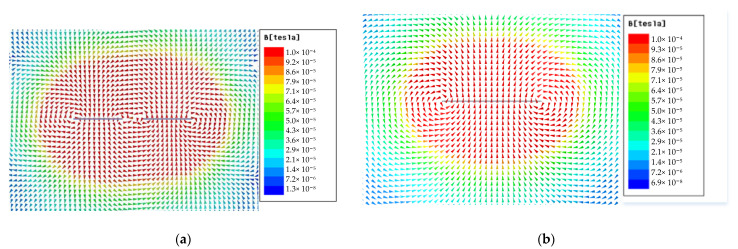
(**a**) Primary field of the x-component transmitting coil; (**b**) primary field of the z-component transmitting coil.

**Figure 5 sensors-22-01648-f005:**
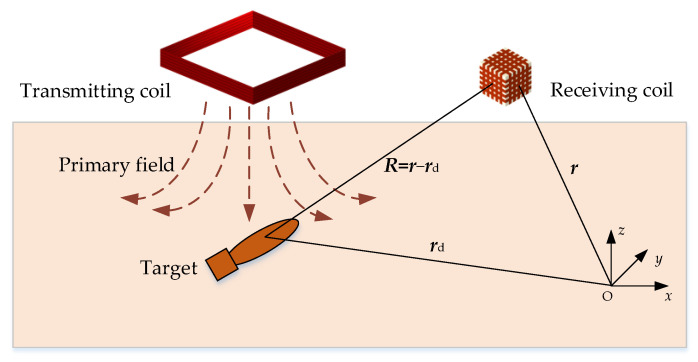
Principle of target detection based on a single dipole model.

**Figure 6 sensors-22-01648-f006:**
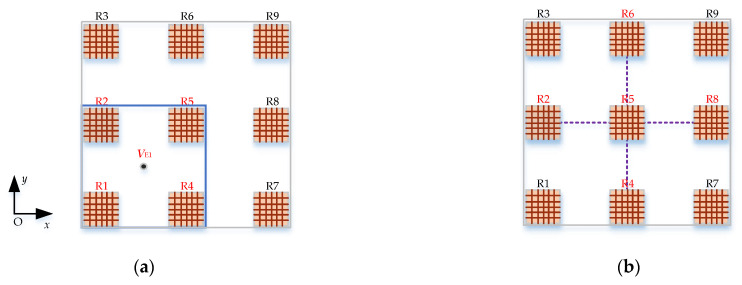
(**a**) Schematic diagram of four-coil tensor position; (**b**) schematic diagram of five-coil tensor position.

**Figure 7 sensors-22-01648-f007:**
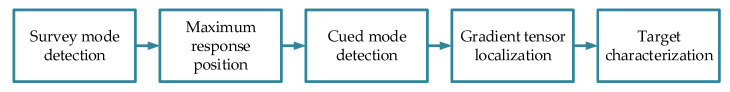
Target detection and data processing.

**Figure 8 sensors-22-01648-f008:**
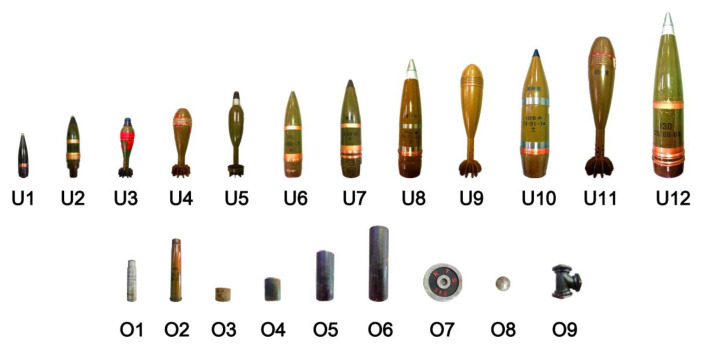
Picture of the targets.

**Figure 9 sensors-22-01648-f009:**
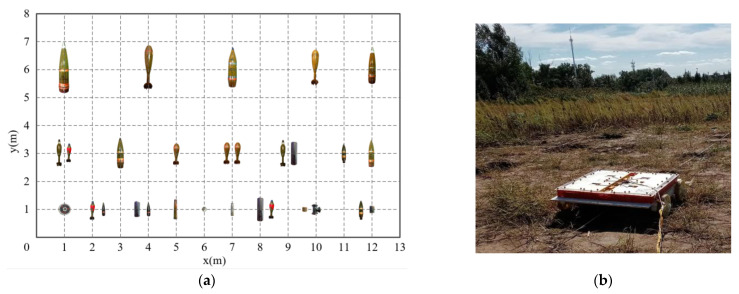
(**a**) Distribution of buried targets in the experiment; (**b**) picture of the field experiment.

**Figure 10 sensors-22-01648-f010:**
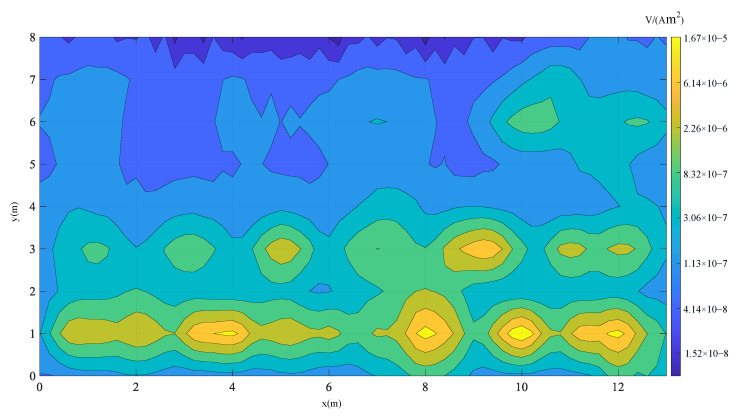
Normalized z component response of the receiving coil R5.

**Figure 11 sensors-22-01648-f011:**
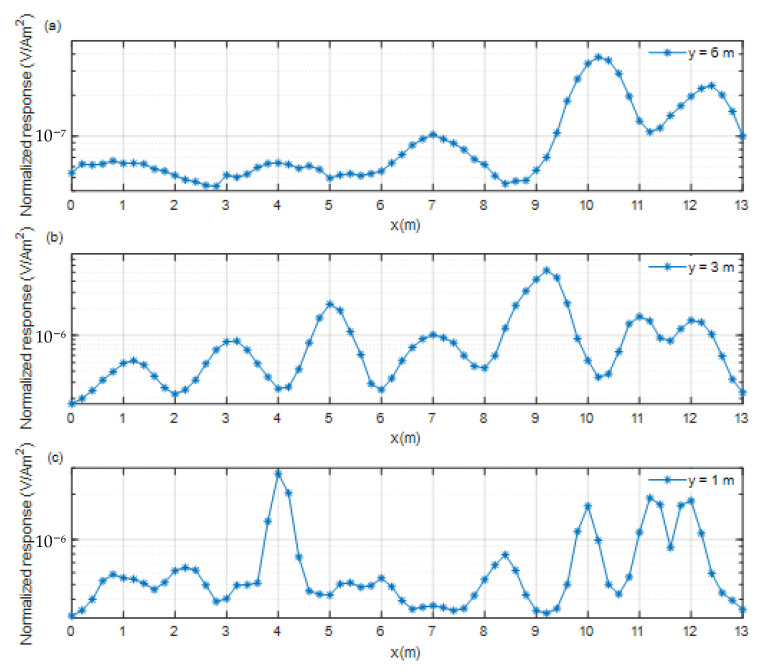
Responses of three survey lines. (**a**) y = 6 m, (**b**) y = 3 m, (**c**) y = 1 m.

**Figure 12 sensors-22-01648-f012:**
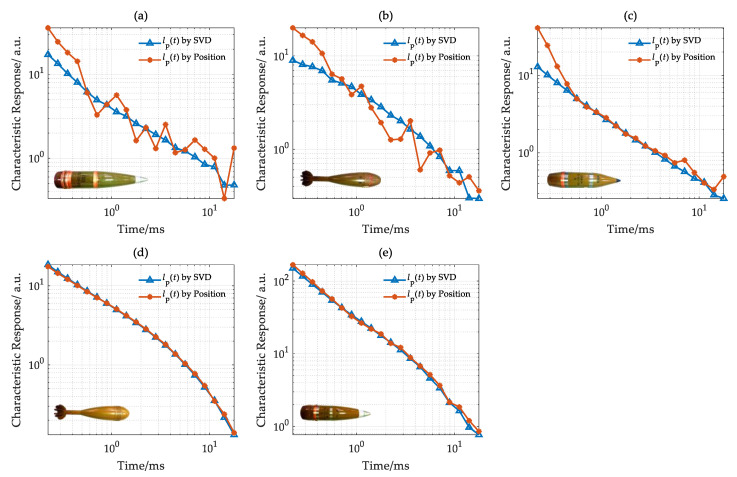
Characteristic response in the survey line with y = 6.0 m. (**a**) U12, (**b**) U11, (**c**) U10, (**d**) U9, (**e**) U8.

**Figure 13 sensors-22-01648-f013:**
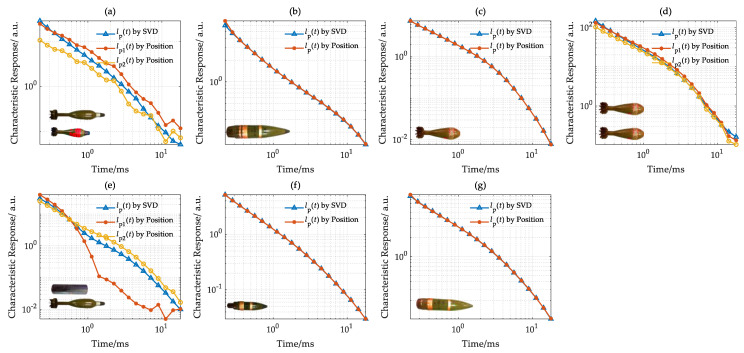
Characteristic response in the survey line with y = 3.0 m. (**a**) U5 and U3, (**b**) U7, (**c**) U4, (**d**) U4, (**e**) O6 and U5, (**f**) U2, (**g**) U6.

**Figure 14 sensors-22-01648-f014:**
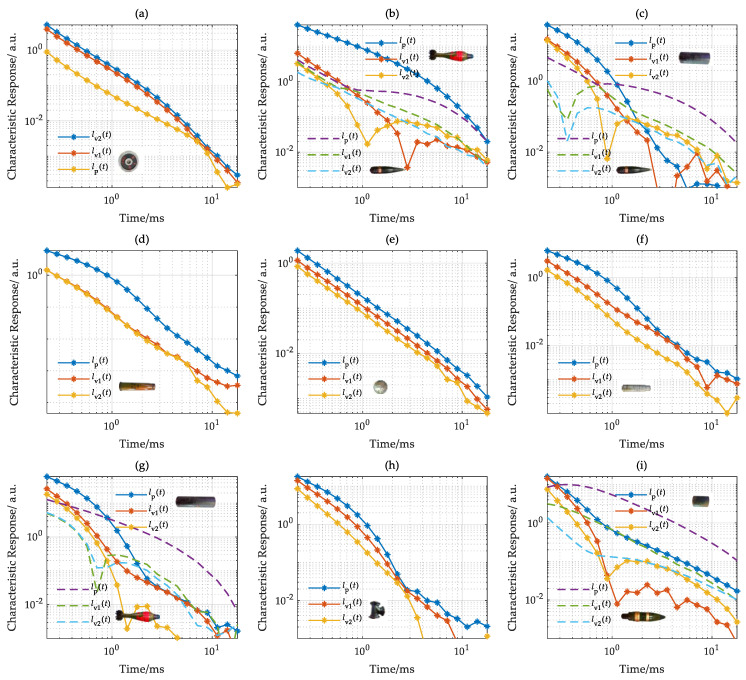
Characteristic response in the survey line with y = 1.0 m. (**a**) O7, (**b**) U3 and U1, (**c**) O5 and U5, (**d**) O2, (**e**) O8, (**f**) O1, (**g**) O6 and U3, (**h**) O9, (**i**) O4 and U2.

**Table 1 sensors-22-01648-t001:** Parameters of the towed sensor.

Coil	Parameter	x	y	z
Transmitting coils	Length (cm)	35 × 85	35 × 85	95 × 95
	Number of turns	20	20	20
	Resistance (Ω)	0.556	0.564	0.468
	Inductance (mH)	1.312	1.316	0.958
Receiving coils	Length (cm)	8.0	8.0	8.0
	Number of turns	480	480	480
	Resistance (Ω)	108.5	111.3	114
	Inductance (mH)	18.49	19.12	19.78
	Resonant frequency (kHz)	231	230	230

**Table 2 sensors-22-01648-t002:** Parameters of UXOs.

Name	U1	U2	U3	U4	U5	U6	U7	U8	U9	U10	U11	U12
Length (cm)	18	26	24	27	34	35	39	51	46	55	58	65
Diameter (mm)	37	57	60	82	75	74	85	100	100	122	120	130
Number	2	2	3	3	2	1	1	1	1	1	1	1

**Table 3 sensors-22-01648-t003:** Parameters of harmless targets.

Name	O1	O2	O3	O4	O5	O6	O7	O8	O9
Length (cm)	16	25	5	10	20	30	2	/	12.5
Diameter (mm)	30	37	75	75	75	75	150	64	/
Number	1	1	1	1	1	2	1	1	1

**Table 4 sensors-22-01648-t004:** Localization results of targets.

Name	True Position (m)	Inverted Position (m)	Error (cm)	DE Algorithm (s)	Proposed Method (s)
U12	(0.90, 0.00, −2.09)	(0.69, −0.25, −2.22)	(−21, −25, −13)	4.02	0.041
U11	(4.00, 0.00, −2.06)	(3.80, −0.24, −2.14)	(−20, −24, −8)	4.39	0.041
U10	(7.03, 0.00, −1.66)	(7.10, 0.03, −1.69)	(7, 3, −3)	4.46	0.040
U9	(10.09, 0.00, −1.26)	(10.20, 0.05, −1.28)	(11, 5, −2)	3.84	0.040
U8	(12.05, 0.00, −1.45)	(11.99, −0.06, −1.40)	(−6, −6, 5)	3.83	0.040

**Table 5 sensors-22-01648-t005:** Localization results of targets.

Name	True Position (m)	Inverted Position (m)	Error (cm)	DE Algorithm (s)	Proposed Method (s)
U5	(0.84, 0.00, −1.03)	(0.94, 0.07, −1.04)	(10, 7, −1)	3.96	0.042
U3	(1.19, 0.00, −1.04)	(1.09, 0.06, −1.11)	(−10, 6, −7)	3.93	0.040
U7	(3.06, 0.00, −1.02)	(3.12, 0.09, −1.07)	(6, 9, −5)	3.94	0.040
U4	(5.02, 0.00, −0.85)	(5.01, −0.01, −0.91)	(−1, −1, −6)	3.92	0.040
U4	(6.85, 0.00, −1.06)	(6.90, 0.06, −1.07)	(5, 6, −1)	3.91	0.041
U4	(7.29, 0.00, −0.80)	(7.21, −0.14, −92)	(−8, −14, −12)	3.84	0.040
U5	(8.77, 0.00, −0.86)	(8.80, 0.04, −0.88)	(3, 4, −2)	3.87	0.041
O6	(9.24, 0.00, −0.72)	(9.25, 0.00, −0.73)	(1, 0, −1)	3.87	0.040
U2	(11.00, 0.00, −0.82)	(10.95, 0.01, −0.86)	(−5, 1, −4)	3.85	0.039
U6	(12.10, 0.00, −0.98)	(12.04, 0.07, −1.02)	(−6, 7, −4)	3.87	0.040

**Table 6 sensors-22-01648-t006:** Localization results of targets.

Name	True Position (m)	Inverted Position (m)	Error (cm)	DE Algorithm (s)	Proposed Method (s)
O7	(1.00, 0.00, −0.46)	(1.02, 0.03, −0.47)	(2, 3, −1)	3.71	0.038
U3	(2.00, 0.00, −0.45)	(2.00, 0.06, −0.50)	(0, 6, −5)	3.81	0.040
U1	(2.40, 0.00, −0.46)	(2.30, 0.03, −0.53)	(−10, 3, −7)	3.87	0.039
O5	(3.60, 0.00, −0.45)	(3.62, 0.03, −0.51)	(2, 3, −6)	3.92	0.040
U1	(4.00, 0.00, −0.49)	(4.01, 0.05, −0.45)	(1, 5, 4)	3.85	0.038
O2	(5.00, 0.00, −0.50)	(4.96, 0.08, −0.52)	(−4, 8, −2)	3.76	0.039
O8	(6.00, 0.00, −0.47)	(6.05, 0.14, −0.49)	(5, 14, −2)	3.88	0.040
O1	(6.98, 0.00, −0.46)	(7.04, 0.10, −0.52)	(6, 10, −6)	3.87	0.040
O6	(8.00, 0.00, −0.46)	(8.00, 0.10, −0.41)	(0, 10, 5)	3.83	0.041
U3	(8.40, 0.00, −0.47)	(8.50, 0.12, −0.48)	(10, 12, −1)	3.81	0.039
O3	(9.60, 0.00, −0.46)	/	/	/	/
O9	(10.00, 0.00, −0.43)	(10.04, 0.11, −0.41)	(4, 11, 2)	3.83	0.038
U2	(11.60, 0.00, −0.42)	(11.61, 0.10, −0.44)	(1, 10, −2)	3.84	0.039
O4	(12.00, 0.00, −0.45)	(11.91, 0.10, −0.52)	(−9, 10, −7)	3.81	0.038

## Data Availability

No new data were created or analyzed in this study. Data sharing is inapplicable to this article.

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
