# Peer review of "Fast Localization and Characterization of Underground Targets with a Towed Transient Electromagnetic Array System"

_sensors, 2022, doi:10.3390/s22041648_

Round 1
Reviewer 1 Report
The paper is well written. The subject is within the scope of the journal and the objective of research is well stated.
The presentation is very good.
Author Response
Dear Reviewer,
We sincerely thanks for your valuable comments. The revised contents are as follows:
1)Moderate English changes required.
Response:
Thank you for your suggestions to improve our manuscript. We have made moderate English corrections.

Reviewer 2 Report
The authors present a towed transient EM array system for UXO detecton and clasiffication based on the magnetic polarizability tensor and gradiometry. They use SVD to obtain the target response without the need for the target location. The approach is verified using field test with 12 buried UXOs and 9 harmless objects.
In general, the paper is well written. I have a few comments, though:
1) I would like to see discussion on the limitations of the field test, particularly with respect to the fact that the test was not blind, i.e. the locations of the targets were known before the test and the scanning was done along the lines where the targets were buried.
2) Although the SNR is mentioned several times as a key factor in the accuracy of the algorithms, no quantitative analysis of the SNR was given. It is not clear what low or high is SNR for the authors. It is not clear if it defined as signal power of each of the receiver channels with respect to its noise power. Also, the noise level of the signal chain is not given. The SNR should be quantified throughout the paper instead of using vague “high” and “low” terms.
3) There is no referenced work in sections 3.1. – 3.3. The dipole model based target detection is developed before in a number of studies, so it would be good to make a difference between already established theory and the authors’ contribution.
4) In section 4.1., there is no data on the depth of the targets whereas the x-y locations are given. Only later, in 4.3. it is clear what the depths are. Perhaps, target depth can be denoted on Fig. 9.a.
5) Please elaborate more line 130-132 in page 4. What is “capacitive energy storage technology”?
6) On line 86 in page 2, the authors state “...a towed array sensor is designed in this paper”. Is it really a fact that this system is presented for the first time in this paper? In addition, there is a lot of previous work done on towed, EM induction sensor arrays, so perhaps a discussion on differences with respect to the state of the art would be welcome.
Author Response
Dear Reviewer,
We sincerely thanks for your valuable comments. The revised contents are as follows:
- I would like to see discussion on the limitations of the field test, particularly with respect to the fact that the test was not blind, i.e. the locations of the targets were known before the test and the scanning was done along the lines where the targets were buried.
Response:
Thank you for your suggestions. The limitations of field test are mainly as follows: First, the size of our sensor is approximately 1m*1m, and the internal of the measurement line spacing is also 1m. Therefore, the sensor can pass through the whole measurement site without omission. Whether the target is directly below or not, the difference in target response is limited, so it will not have a significant impact on the detection results; Second, there is a certain distance between the two groups of targets buried along the measurement line. Therefore, it can be considered that the detection for each group of targets is carried out independently without interference with each other. These targets are buried along the measurement line to improve the efficiency of the whole experiment.
- Although the SNR is mentioned several times as a key factor in the accuracy of the algorithms, no quantitative analysis of the SNR was given. It is not clear what low or high is SNR for the authors. It is not clear if it defined as signal power of each of the receiver channels with respect to its noise power. Also, the noise level of the signal chain is not given. The SNR should be quantified throughout the paper instead of using vague “high” and “low” terms.
Response:
Thank you for your suggestions. In transient electromagnetic detection, the measured response includes target response, environmental noise, and noise of the sensor itself. The noise of the sensor itself is very low, and the noise in the response mainly refers to environmental noise. The ambient noise can usually be suppressed by transient sampling and superposition. The target response is constantly changing along the measurement line, and the response of each time channel is different. The ambient noise also fluctuates greatly during the whole measurement time, so it is difficult to describe the signal quality with a single noise level, such as power spectrum. Generally, the SNR level can be roughly judged according to the signal amplitude and the smoothness of the response curve, and a relatively high and low SNR judgment can be given.
- There is no referenced work in sections 3.1. – 3.3. The dipole model based on target detection is developed before in a number of studies, so it would be good to make a difference between already established theory and the authors’ contribution.
Response:
Thank you for your suggestion. In sections 3.1, we have added reference (17, 26) to make a difference between already established theory and the authors’ contribution.
- In section 4.1., there is no data on the depth of the targets whereas the x-y locations are given. Only later, in 4.3. it is clear what the depths are. Perhaps, target depth can be denoted on Fig. 9.a.
Response:
Thank you for your suggestion. In order to better show the errors between the real depth of the target and the estimated depth, we give the real depths of the target in tables 4 to 6.
- Please elaborate more line 130-132 in page 4. What is “capacitive energy storage technology”?
Response:
Figure 1 (a) shows the transmitter without an energy storage capacitor. When S1 and S4 are on, the current in the transmitting coil rises exponentially, with a time constant of L / R. For the inductance of transmitting coil is about 1 mH, and the resistance is 0.5 ohm, the time constant is sbout 2 ms. When the system operates at 125 Hz, the pulse width of the emission current is only 2 ms, only the time of one time constant. Therefore, the current amplitude is low during the emission time, as shown in Figure 1(c). Figure 1(b) shows the transmitter with an energy storage capacitor. When the MOSFET S1 and S4 are turned off, the current flowing in L will flow to C through S2 and S3. That is, the energy in transmitting coil L is storage in capacitor C. The voltage on the capacitor after charging is between 150-200v. When S4 and S2 are turned on, the capacitor is directly charged into the coil. Due to the high voltage on the capacitor, the charging speed is fast, and the front edge of the emission current will rise rapidly, so as to ensure that the current amplitude is significantly increased during the whole emission period, as shown in Figure 1(d). Thanks again for your suggestion to improve our paper.
Figure 1. (a) H-bridge without energy storage capacitor; (b) H-bridge circuit with energy storage capacitor, C is energy storage capacitor; (c) Current without energy storage capacitor; (d) Current with energy storage capacitor.
- On line 86 in page 2, the authors state “...a towed array sensor is designed in this paper”. Is it really a fact that this system is presented for the first time in this paper? In addition, there is a lot of previous work done on towed, EM induction sensor arrays, so perhaps a discussion on differences with respect to the state of the art would be welcome.
Response:
Thank you for your suggestion. At present, there are mainly three types of towed transient electromagnetic sensors: Metal-Mapper (MM), Berkeley UXO Discriminator (BUD), and Time-domain Electromagnetic Multi-sensor Tower Array Detection System (TEMTADS). MM and BUD are three-component structures in space, and their XY direction coils are all upright rectangular transmitting coils. TEMTADS is a flat square matrix structure with low height and good stability. Starting from this structure, we design the three-component planar structure of this paper. Thanks again for your suggestion to improve our paper.

Reviewer 3 Report
1. In the abstract, the authors make multiple inferences to the "traditional algorithm" and "traditional method", which is rather vague and confusing. If the "traditional" refers to methods that do not rely on magnetic gradient tensor, but rather on the standard induced dipole model where the target localization problem is nonlinear and therefore requires more complex iterative inversion algorithms, then this should be somehow made clear right from the beginning of the paper.
2. Introduction, Page 2. "These algorithms usually require dozens of or thousands of iterations, resulting in a time-consuming and low-efficiency process."
There is a big difference in computational time and efficiency between algorithms requiring dozens of iterations (e.g. Gauss-Newton, GN) and those requiring thousands of iterations (e.g. DE), so making such general claims should be avoided. From the reviewer's experience, GN-based inversion algorithms can be quite effective in TEM applications, provided that good initial solution can be obtained and efficient forward models are used. I think that the question "gradient tensor based" vs. "traditional" (iterative) methods is not only about computational efficiency, but also about dealing with the nonlinearity of the original inversion problem and the related trade-offs. The authors should take this into account and when comparing and discussing the two approaches.
3. Introduction, Page 2. "The localization accuracy... is mainly affected by .. and equivalent error of the magnetic gradient tensor."
Please explain what is meant by the "equivalent error of the magnetic gradient tensor"? Is the numerical error related to the calculation of gradient field by combining measurements from different sensors? If so, perhaps some other term would be more appropriate.
4. Section 3.2. Please provide some coarse quantitative assessment on how the distance between the coils used to determine the gradient tensor affects the spatial resolution.
5. In relation to expression (13), it is not clear how the evolution of the characteristic matrix L(t) over time instances Nt was obtained from SVD of measurement matrix V which has a dimension of 27x3? Have the authors performed one SVD for each time instance or have they used a joint SVD algorithm to simultaneously decompose multiple matrices V over different time instances using a shared set of singular vectors? This needs to be explained.
6. Page 10. In relation to the distribution of buried targets in the experiment, Fig. 9, burial depths are not defined?
7. Page 12. "As shown in Table 4,..., the SNR of the horizontal component response is extremely low."
SNR values are not shown in Table 4?!
8. In relation to Table 4, if the proposed method is being compared against DE algorithm in terms of execution time, it would be also fair to compare them in terms of position estimation accuracy. This also goes for Tables 5 and 6.
Author Response
Dear Reviewer,
We sincerely thanks for your valuable comments. The revised contents are as follows:
- In the abstract, the authors make multiple inferences to the "traditional algorithm" and "traditional method", which is rather vague and confusing. If the "traditional" refers to methods that do not rely on magnetic gradient tensor, but rather on the standard induced dipole model where the target localization problem is nonlinear and therefore requires more complex iterative inversion algorithms, then this should be somehow made clear right from the beginning of the paper.
Response:
Thank you for your suggestion. In the abstract, traditional algorithms usually refer to some iterative algorithms, such as differential evolution and Gauss-Newton algorithm. These algorithms all use an iterative method to deal with the nonlinear inversion problem of dipole positioning. We modify the third line of the abstract "The traditional algorithm usually iterates tens to thousands of times" to "The traditional algorithms, such as differential evolution or Gauss-Newton algorithms, usually iterate tens to thousands of times to locate the underground target ". Thanks again for your suggestion to improve our paper.
- Introduction, Page 2. "These algorithms usually require dozens of or thousands of iterations, resulting in a time-consuming and low-efficiency process." There is a big difference in computational time and efficiency between algorithms requiring dozens of iterations (e.g. Gauss-Newton, GN) and those requiring thousands of iterations (e.g. DE), so making such general claims should be avoided. From the reviewer's experience, GN-based inversion algorithms can be quite effective in TEM applications, provided that good initial solution can be obtained and efficient forward models are used. I think that the question "gradient tensor based" vs. "traditional" (iterative) methods is not only about computational efficiency, but also about dealing with the nonlinearity of the original inversion problem and the related trade-offs. The authors should take this into account and when comparing and discussing the two approaches.
Response:
Thank you for your suggestion. It is true that there are some problems with my statement here. Through iterations, traditional algorithms, such as differential evolution and Gauss Newton algorithms, can obtain better target localization results based on dipole model. In view of the situation of the dipole forward model, this paper uses the magnetic gradient tensor method to transform the nonlinear problem into a linear problem, thereby improving the processing efficiency. We have added explanations on pages 72 to 74 of this paper. Thanks again for your suggestion to improve our paper.
- Introduction, Page 2. "The localization accuracy... is mainly affected by. and equivalent error of the magnetic gradient tensor." Please explain what is meant by the "equivalent error of the magnetic gradient tensor"? Is the numerical error related to the calculation of gradient field by combining measurements from different sensors? If so, perhaps some other term would be more appropriate.
Response:
Thank you for your suggestion. The equivalent error of magnetic gradient tensor refers to the error introduced by the magnetic gradient tensor G calculated for different sensor structures. When the target is shallow, as shown in Figure 1, the error of magnetic gradient tensor G can be reduced by using four close sensors R1, R2, R4, and R5. Thank you again for your suggestion. We have adjusted the terminology, as shown in Figure 1.
Figure 1. Schematic diagram of four-coil tensor position.
- Section 3.2. Please provide some coarse quantitative assessment on how the distance between the coils used to determine the gradient tensor affects the spatial resolution.
Response:
Thank you for your suggestion. The distance between adjacent coils of the towed system designed in this paper is 20 cm. When the target distance is shallow, that is when the target depth is about 50 cm, it is suitable to use the magnetic gradient tensor constructed in Figure 2(a) for target positioning. When the target depth is deep, that is, when the target depth is greater than 1 m, it is suitable to use the magnetic gradient tensor constructed in Figure 2(b) for target positioning. At this time, the coil distance for calculating the difference is 40 cm. In general, the appropriate sensor structure can be determined according to the relation of the target depth and coil distance. Overall, when the target depth is twice the sensor size, it can guarantee the accuracy of magnetic gradient tensor localization.
|
(a) |
(b) |
Figure 2. (a) Schematic diagram of four-coil tensor position; (b) schematic diagram of five-coil tensor position.
- In relation to expression (13), it is not clear how the evolution of the characteristic matrix L(t) over time instances Nt was obtained from SVD of measurement matrix V which has a dimension of 27x3? Have the authors performed one SVD for each time instance or have they used a joint SVD algorithm to simultaneously decompose multiple matrices V over different time instances using a shared set of singular vectors? This needs to be explained.
Response:
Thank you for your suggestion. The system designed in this paper is composed of three transmitting coils and nine three-component receiving coils. In the field test, three transmitting coils are used to excite the underground target in turn, so the size of the received response V is 27×3. A joint SVD algorithm to simultaneously decompose multiple matrices V. It is true that we have performed SVD processing on the matrix of each time channel to obtain the decay law of the electromagnetic characteristics of the target over time.
- Page 10. In relation to the distribution of buried targets in the experiment, Fig. 9, burial depths are not defined?
Response:
Thank you for your suggestion. The depths of these targets are all different, and we give the true depth of the target in Tables 4-6 and compare it with the inversion results.
- Page 12. "As shown in Table 4,..., the SNR of the horizontal component response is extremely low." SNR values are not shown in Table 4?!
Response:
Thank you for your suggestion. We have revised the statement. It can be seen from Table 4 that the error of the horizontal positioning is relatively large. This is because the target is directly below the sensor and is more than 2.0 m in depth. Therefore, the horizontal component of the target response received by the receiver coils directly above the target is very small. Thanks again for your suggestion to improve our paper.
- In relation to Table 4, if the proposed method is being compared against DE algorithm in terms of execution time, it would be also fair to compare them in terms of position estimation accuracy. This also goes for Tables 5 and 6.
Response:
Thank you for your suggestion. Compared with the iterative algorithm, such as DE algorithm, the proposed method can significantly improve the running time on the premise of ensuring the positioning accuracy. The proposed method needs approximately 40 ms, only 1% of the traditional one. Thanks again for your suggestion to improve our paper.

Reviewer 4 Report
This paper does not provide any new valuable scientific results. all of the presented results and conclusions can be found here:
Shubitidze F. A Complex Approach to UXO Discrimination: Combining Advanced EMI Forward and Statistical Signal
Processing. 2012.
Namely, Shubitidze in his report describes:
the magnetic field gradient approach for locating targets, as well as multi-static TEM sensor, SVM technique and multi-targets inversion and classifications.
Authors should state what are new contributions here.
Author Response
Dear Reviewer,
We sincerely thanks for your valuable comments. The revised contents are as follows:
- This paper does not provide any new valuable scientific results. all of the presented results and conclusions can be found here: Shubitidze F. A Complex Approach to UXO Discrimination: Combining Advanced EMI Forward and Statistical Signal Processing. 2012. Namely, Shubitidze in his report describes: the magnetic field gradient approach for locating targets, as well as multi-static TEM sensor, SVM technique and multi-targets inversion and classifications.
Authors should state what are new contributions here.
Response:
We sincerely thanks for your valuable comments. Our contributions are stated as follows:
- The gradient search method in reference [Shubitidze F. A Complex Approach to UXO Discrimination: Combining Advanced EMI Forward and Statistical Signal 2012.] is an iterative algorithm, which is used to solve the optimal solution problem of the target location under the least square constraint. It is different from the magnetic gradient tensor algorithm constructed in this paper;
- In this paper, the magnetic gradient tensor method in magnetic detection [A Closed-Form Formula for Magnetic Dipole Localization by Measurement of Its Magnetic Field and Spatial Gradients] is firstly applied to TEM UXO detection without iteration;
- In this paper, the magnetic gradient tensor is used to locate the target, which has fast localization speed (about 40 ms) and high localization accuracy (the depth error is no more than 13 cm). Besides, two types of magnetic gradient tensors, rectangular and cross-shaped, are constructed according to the 3×3 sensor array, which overcomes the problem of large equivalent error in inversion.
The final location results show that the proposed method can effectively achieve fast and accurate target location, and provide a new idea and method for fast detection and location of UXOs. Thanks again for your suggestion.